# Differences in Antiphospholipid Antibody Profile between Patients with Obstetric and Thrombotic Antiphospholipid Syndrome

**DOI:** 10.3390/ijms232112819

**Published:** 2022-10-24

**Authors:** Ariadna Anunciación-Llunell, Cándido Muñoz, Dirk Roggenbuck, Stefano Frasca, Josep Pardos-Gea, Enrique Esteve-Valverde, Jaume Alijotas-Reig, Francesc Miró-Mur

**Affiliations:** 1Systemic Autoimmune Diseases Research Unit, Vall d’Hebron Institut de Recerca (VHIR), 08035 Barcelona, Catalonia, Spain; 2Centre for Rheumatology Research, University College of London, London WC1E 6JF, UK; 3Faculty of Health Sciences Brandenburg, Brandenburg University of Technology Cottbus-Senftenberg, 01968 Senftenberg, Germany; 4GA Generic Assays GmbH, 15827 Dahlewitz, Germany; 5Systemic Autoimmune Diseases Unit, Department of Internal Medicine, Hospital Universitari Vall d’Hebron (HUVH), 08035 Barcelona, Catalonia, Spain; 6Systemic Autoimmune Diseases Unit, Department of Internal Medicine, Hospital Universitari Parc Taulí, 08208 Sabadell, Catalonia, Spain; 7Department of Medicine, Faculty of Medicine, Universitat Autònoma de Barcelona, 08035 Barcelona, Catalonia, Spain

**Keywords:** antiphospholipid antibodies, thrombotic antiphospholipid syndrome, obstetric antiphospholipid syndrome, non-criteria antiphospholipid antibody, line Immunoassay

## Abstract

Antiphospholipid syndrome (APS) is a systemic autoimmune condition characterised by the presence of antiphospholipid antibodies (aPL) associated with vascular thrombosis and/or pregnancy complications. In a cohort of 74 yet diagnosed APS individuals fulfilling Sydney laboratory criteria (twice positive for lupus anticoagulant, anticardiolipin, aCL, and/or anti-β2glycoprotein I, aβ2GPI), 33 out of 74 were obstetric APS (OAPS) and 41 thrombotic APS (TAPS) patients. 39% of TAPS patients were women. Although aPL detection was persistent, we observed an oscillatory aPL positivity in 56.7% and a transient seroconversion in 32.4% of APS patients at enrolment. Thus, we tested their sera in a line immunoassay that simultaneously detected IgG or IgM for criteria (aCL and aβ2GPI) and non-criteria (anti-phosphatidylserine, aPS; anti-phosphatidic acid, aPA; anti-phosphatidylinositol, aPI; anti-annexin 5, aA5; anti-prothrombin, aPT; anti-phosphatidylethanolamine; anti-phosphatidylglycerol, and anti-phosphatidylcholine) aPL. OAPS and TAPS patients displayed different but overlapping clusters based on their aPL reactivities. Specifically, while OAPS patients showed higher aPA, aPS, aA5, aβ2GPI and aPT IgM levels than TAPS patients, the latter displayed higher reactivity in aCL, aPI and aA5 IgG. Eventually, with a cut-off of the 99^th^ percentile established from a population of 79 healthy donors, TAPS patients significantly tested more positive for aCL and aA5 IgG than OAPS patients, who tested more positive for aPA, aPS and aβ2GPI IgM. Transiently seronegative APS patients showed non-criteria aPL positivity twice in sera obtained 3 months apart. Overall, our data show that APS patients presented clusters of aPL that define different profiles between OAPS and TAPS, and persistent non-criteria aPL positivity was observed in those who are transiently seronegative.

## 1. Introduction

Antiphospholipid syndrome (APS) is an autoimmune disease characterised by vascular thrombosis, various obstetrical adverse events and persistent antiphospholipid antibodies (aPL). The conventionally accepted aPL in terms of classification criteria (Sydney criteria) include lupus anticoagulant (LA), IgG/IgM anticardiolipin antibodies (aCL) and IgG/IgM antibodies against β2-glycoprotein I (aβ2GPI) [1,2]. Patients with clinical obstetric and thrombotic APS features but without detectable criteria aPL are defined as seronegative or non-criteria APS patients [3]. Since then, methodological approaches to detect new antigenic targets have been developed and several non-criteria aPL can be detected in patients with clinical APS features [4,5]. The group of non-criteria aPL encompasses anti-phosphatidylethanolamine (aPE), anti-phosphatidylserine/prothrombin (aPS/PT) complex, anti-vimentin, and anti-annexin 5 (aA5) among others [6]. The detection of non-criteria aPL aids in the serological diagnosis of seronegative APS and warrants a similar therapeutic management as seropositive APS [7]. The prevalence of these persistent non-criteria aPL is notorious in distinct thrombotic APS (TAPS) subsets, as for instance in 10–15% of APS patients with unexplained venous thrombosis [8]. In preeclampsia (PE) the occurrence of multiple aPL encompassing non-criteria aPL was associated with severe PE disease [9]. However, there are few studies investigating the link between the occurrence of aPL and obstetrical or thrombotic outcomes.

As there is accumulating evidence that some non-criteria APS patients could be persistently negative for criteria aPL, the detection of non-criteria aPL appears to be essential for their diagnosis. Moreover, changes in aPL titres during pregnancy sometimes cause loss of aPL positivity [10]. In the last decade, several studies have reported obstetric patients suffering from seronegative-APS for whom non-criteria aPL might be present [11]. A recent retrospective study reported that seronegative APS is rather more obstetrical than thrombotic phenotypes [7]. The cumulative incidence of adverse obstetrical events was similar in seronegative and seropositive APS patients, although higher rates of intrauterine deaths, PE, and lower live birth term were observed in seropositive APS [7]. When comparing obstetric APS (OAPS) and non-criteria OAPS, and both receiving the same treatment, similar foetal–maternal outcomes were observed [12]. In addition, the use of methodologies that simultaneously screen for multiple criteria and non-criteria aPL help in the laboratory diagnosis of APS [13,14] as well as in defining aPL profiles [15] and clinical APS phenotypes [16].

Furthermore, signalling pathways at the intersections of coagulation and innate immune signalling distinct from those induced by LPS could be activated by aPL through endothelial protein C receptor (EPCR) [17]. Whether this molecular signalling depends on cellular or molecular specificity is still elusive, but aPL signalling by only lipid-reactive antibodies or by aβ2GPI triggers the EPCR pathway both in immune cells and in trophoblasts [17]. Thus, different aPL could induce common molecular pathways leading to different cellular signals and activation.

In contrast to patients with OAPS, only patients with thrombotic manifestations carry an increased risk of subclinical atherosclerosis [18]. Thus, distinctive pathogenic mechanisms may be responsible for the two outcome. In vivo models of foetal loss suggested that aPL effects could be mediated by acute placental inflammation. However, histopathological examination of APS placentae did not support a widespread inflammatory signature [19]. aPL are thought to recognise their antigens on placental tissues, inhibit the growth and differentiation of trophoblast cells and, eventually cause defective placentation on the basis of thrombogenic mechanisms [20]. However, recurrent pregnancy complications associated with aPL cannot be explained solely by thrombosis and alternative pathogenic mechanisms must be considered. Genic variants related to blood coagulation pathways and the immune responses were found to be associated with thrombotic APS and IgG purified from both types of pathologies was identified to modulate genes related to cell responses to stress, cell adhesion and extracellular matrix, MAPK signalling modulation and cell interactions [21]. These different mechanisms could be initiated by aPL having unique antigen specificity, and characterisation of the molecular basis of pathogenic mechanisms may help in the diagnosis and choice of therapy options.

The current classification criteria limit the laboratory parameters to three groups of aPL: LA, aCL and aβ2GPI. However, the optimisation of laboratory diagnosis and risk stratification opens the field towards the so-called non-criteria aPL. In this study, we explore the differences in aPL reactivity between OAPS and TAPS patients by assessing criteria and non-criteria aPL with a multiplex line immune assay (LIA). This will provide further evidence for the presence of diverse pathological mechanisms in both clinical phenotypes of APS based on the corresponding aPL profile.

## 2. Results

### 2.1. Demographic and Laboratory Data of APS Patients

A cohort of 74 diagnosed APS patients fulfilling the laboratory Sydney criteria, twice positive for LA, aCL and/or aβ2GPI, were divided according to their clinical criteria into obstetric (OAPS, *n* = 33) and thrombotic (TAPS, *n* = 41) APS patients. Differences in demographic and clinical data between both subsets, but not in laboratory category were observed (Table 1). Thrombotic patients were older than obstetric ones, and only 39% of patients were women in TAPS. The thrombotic APS patients displayed higher body mass indices and showed higher percentages of arterial hypertension or dyslipidaemia. In spite of these differences in demographic and clinical features, both subsets of APS had similar frequencies of patients belonging to one of the four laboratory categories (Table 1). However, significant differences were observed for the number of patients being persistently positive for any aPL. Whereas TAPS patients showed a higher frequency of persistent aPL positivity than OAPS patients (23/41 vs. 9/24, *p* = 0.018), the latter showed a more oscillatory profile (Table 1).

### 2.2. Criteria and Non-Criteria aPL Screening in Healthy and APS Individuals

Given the lack of significant differences in the laboratory categories between the OAPS and TAPS groups and the relatively high percentage of APS patients without criteria aPL at enrolment in both subsets (TAPS: 34.1%, OAPS: 30.3%), we searched for the occurrence of non-criteria aPL in these patients. To this end, we detected 10 different aPL IgG or IgM by LIA (GA Generic Assays GmbH, Dahlewtiz, Germany) in both APS subsets and in 79 age- and gender-matched healthy donors. First, we determined individual cut-offs for each aPL using the group of healthy donors (Figure 1). The highest percentiles assessed from healthy donors for each aPL are depicted in Table 2. Some aPL, such as anti-phosphatidylcholine (aPC), showed little reactivity to both IgG and IgM, which resulted in low cut-off values.

To check for the linearity of aPL detection by LIA and the location of the cut-off on each aPL curve, we performed a serial dilution of a serum from an APS patient who was strongly positive for aCL IgG and aβ2GPI IgG by a commercial chemiluminescent assay. This APS patient also showed reactivity to other non-criteria aPL by LIA (Figure 2a). The established cut-off values (99th percentile) were in the linear parts of each corresponding aPL curve (Figure 2b, red line). These data show that the established cut-offs can readily be used to discriminate aPL positivity.

### 2.3. aPL Association in APS Patients

Since we simultaneously detected IgG and IgM isotypes for 10 different aPL, we investigated associations between the distinct aPL (Figure 3). We found several clusters of aPL with high correlations, such as the cluster for anti-phosphatidylserine (aPS) IgM, aCL IgM, and anti-phosphatidic acid (aPA) IgM. A strong correlation was also observed for aCL IgG, aPA IgG, anti-phosphatidylinositol (aPI) IgG and aPS IgG. Moreover, aPE IgG was positively associated with anti-phosphatidylglycerol (aPG) IgG. These results clearly indicated distinct expression profiles of several aPL in APS patients.

### 2.4. Detection of aPL Profiles in TAPS and OAPS Patients

In order to identify differences in the aPL profiles between TAPS and OAPS patients, we performed a multivariate analysis by principal component analysis (PCA) from the data of aPL reactivity. We sought to identify unique aPL profiles and which aPL reactivity contributed most to this analysis. Accordingly, aPS, aPA, aCL, aPI and aPG IgG reactivities mainly contributed to Dimension 1, whereas Dimension 2 was determined by aPS, aCL, aPA and aβ2GPI IgM reactivities (Figure 4a). PCA clearly distinguished IgG from IgM aPL (Figure 4b) and showed a different but overlapping clustering of OAPS and TAPS patients (Figure 4c).

### 2.5. Individual aPL Reactivity in TAPS and OAPS

Our data suggest that OAPS and TAPS patients are characterised by different aPL profiles. To identify which aPL were differently expressed in TAPS and in OAPS patients, we compared the aPL levels in both phenotypes (Figure 5a). We observed that aPA, aPS, aA5, aβ2GPI and aPT IgM levels were significantly higher in OAPS patients, whereas aCL, aPI and aA5 IgG levels were higher in TAPS patients (Figure 5b). Thus, the different aPL profiles displayed by OAPS and TAPS patients encompassed both criteria and non-criteria aPL.

### 2.6. aPL Positivity in TAPS and OAPS Patients

Whether this aPL reactivity could be translated to a different aPL positivity between TAPS and OAPS patients was assessed thanks to the cut-off value of the 99th percentile described in Table 2. TAPS patients significantly tested more positive for aCL and aA5 IgG than OAPS patients, who tested more positive for aPA, aPS and aβ2GPI IgM (Table 3). We can conclude that OAPS patients presented a set of aPL positivity that was different from that displayed by TAPS patients.

### 2.7. Non-Criteria aPL Positivity in Transient Seronegative TAPS and OAPS Patients

We previously mentioned that some OAPS and TAPS patients were seronegative at the moment of enrolment. We analysed these transient seronegative APS patients (10 OAPS and 14 TAPS) for the presence of non-criteria aPL (Table 4). The number of APS patients who resulted positive for any of the non-criteria aPL tested in the LIA was 6 out of 10 (60.0%) in OAPS and 9 out of 14 (64.3%) in TAPS patients. Moreover, we drew blood from these patients in a lapse of time greater than 12 weeks and tested their sera again by LIA. The number of patients with twice medium (>95th percentile) or high (>99th percentile) titers for the same non-criteria aPL was two (20.0%) in OAPS and eight (57.1%) in TAPS. There were no differences in the positivity of non-criteria aPL between seronegative OAPS and TAPS (Table 4). The non-criteria antibodies that tested twice positive in the seronegative OAPS were aA5, aPE, aPG, and aPA. We found an enlarged list of the non-criteria aPL that tested twice positive in seronegative TAPS: aA5, aPE, aPG, aPA, aPI, aPS and aPT.

Since three TAPS patients showed twice aPS and/or aPT positivity, we decided to test all the seronegative APS patients for aPS/PT by ELISA (Aeskulisa Serine-Prothrombin-GM, AESKU.DIAGNOSTICS, Wendelsheim, Germany). We observed that none of the seronegative OAPS patients yielded positive for aPS/PT. The two seronegative TAPS patients who were positive for aPT by LIA also tested positive for aPS/PT by ELISA. A third TAPS patient with twice aPS positivity resulted in medium positivity by aPS/PT (Figure 6).

## 3. Discussion

Our results pointed out the fact that a distinctive set of aPL is detected in OAPS and TAPS patients, suggesting a different profile of aPL in both clinical phenotypes. The diagnosis of APS relies on the detection of heterogeneous aPL and currently only criteria aPL are being accepted for APS classification and, subsequently, for diagnosis. Moreover, in the interpretation of aPL results, antibody profiles help in identifying patients at risk of thrombosis [22]. The non-criteria aPL may be useful in patients with incomplete antibody profiles to confirm or exclude the increased risk profile [23]. The cumulative incidence of adverse obstetrical events was significantly improved in treated versus untreated seronegative APS. These results suggest that we could improve our clinical practice with a better understanding of non-criteria APS patients. Large-scale histological studies found that placental infarction, impaired spiral artery remodelling, decidua inflammation, and the deposition of complement split products were the most common features in the placentas of aPL-positive women. These pathological manifestations suggest the role of angiogenic and inflammatory factors in the pathological process of the disease [24]. aPLs can target β2GPI expressed on the cell surface of trophoblasts, leading to the enhancement of trophoblast secretion of pro-inflammatory cytokines and chemokines via the activation of the Toll-like Receptor 4 (TLR4)/MyD88 pathway. The presence of other aPL could induce alternative molecular pathways resulting in similar pathological mechanisms. Evidence of the existence of alternative pathways could be found in the TNF signalling. TNF-α is a critical effector in aPL-related placental injury and further miscarriage [25]. Although these OAPS patients are treated with the standard of care, some of them are refractory to the treatment and additional therapies are required to improve the aPL-related poor obstetric outcomes [26].

Although the Sydney criteria were developed for classification purposes, the laboratory parameters are used as diagnostic criteria. However, clinical outcomes of APS patients are often not related to criteria aPL detection, suggesting the need for improvement in reliable aPL assays. Non-criteria aPL could help to further define these diagnostic laboratory criteria that are still prone to discussion and need for regular updates. In this study, we have used the LIA technique manufactured by Generic Assays to report the presence of either criteria and non-criteria aPL in obstetric and thrombotic APS patients. Thus, we are describing the profile of aPL (including criteria and non-criteria) in both phenotypes of APS patients rather than searching for non-criteria aPL in seronegative patients. All APS patients included in this study were diagnosed by fitting the laboratory and the clinical Sydney criteria [1]. Nevertheless, aPL titres are not permanently stable, and their levels and even the aPL positivity could change. We observed that around 32% of APS patients were seronegative for the conventional aPL by the time that the sera for the LIA screening was collected. In agreement with [27], indiscriminate testing is strongly discouraged to avoid incidental findings and, consequently, we only tested for aPL in the context of APS. A limitation of our study is that the LIA did not include the detection of the outmost non-criteria aPL such as the anti-domain I of β2GPI protein and the aPS/PT complex antibodies. aPS/PT have been recently detected in patients with either thrombotic events or pregnancy complications [28], and the anti-domain I of β2GPI was described as pathogenic in APS-related manifestations [29,30].

Common pathways of aPL signalling in OAPS and TAPS would be the complement activation. Studies on murine models of APS have shown that C3 and C5 complement components are crucial for mediating aPL-induced thrombosis in mice [31]; in humans, the placenta from OAPS women exhibited high levels of C4a and C3b deposition, which indicated the possibility of complement activation in the pathogenesis of pregnancy complications in OAPS women [32]. aPL could trigger the activation of complement, generating the active components C3a, C3b, and C5a and ultimately leading to the generation of the membrane attack complex (MAC) [33]. In addition, the binding of these ligands to their receptors on target cells could induce TF release by those cells and results in the formation of thrombosis [34]. Nevertheless, as to which aPL induce MAC or TF should be explored.

New technical approaches to aPL profiling have been reported, which elucidate the possibility of aPL profiling for the diagnosis of APS [15]. Besides the LIA, bead-based multiplex techniques to detect several aPL in the same analysis might represent an improvement in the quality of routine laboratory measurements. The different aPL profiles described by our study in OAPS and TAPS patients indicates a more striking underlying immune response against phospholipid structures or corresponding cofactors in these patients, consistent with the conclusion that activation of the immune system is different in those patients who experience obstetrical morbidities compared to those patients who experience thrombosis as clinical manifestations of APS. aPL profiling assists in the differentiation of clinical phenotypes in APS patients. Different studies have reported that the LIA could distinguish APS apart from infectious non-APS patients and could also predict the thrombotic risk in APS patients [14,22].

The clinical meaning of non-criteria antibodies is still being debated, because in the APS diagnosis, triple-positive consensus criteria aPL were more powerful to predict APS early than single-positive or non-criteria aPL positivity [35]. However, the dynamics of aPL levels is also a matter of debate and many APS patients showed an oscillatory aPL positivity. Our data described that thrombotic APS patients who transiently tested seronegative for criteria aPL could be detected as being twice positive for aPS and/or aPT by two different techniques (aPS, aPT by LIA, and aPS/PT by ELISA). Consequently, the reliability of APS diagnosis would improve with testing more aPL.

Principal component analysis based on aPL reactivity clearly discriminated IgG from IgM aPL. We observed that TAPS patients overlaid on IgG area whereas OAPS did on IgM. This observation associated with the finding that whereas thrombotic APS displayed more IgG aPL reactivities than obstetric APS, IgM aPL reactivities were more prevalent in OAPS. The idea that IgM isotypes are more prevalent in gestational than thrombotic APS, and vice versa, was also observed recently in a study describing the prevalence of circulating β2GPI-aβ2GPI immune complexes [36].

A limitation of the study is the relatively small number of OAPS and TAPS patients. Nevertheless, we tested multiplex LIA in a total of 153 individuals, where 79 were healthy donors and 74 were APS patients. Data from healthy donors were used to define the cut-offs for each aPL in the multiplex assay. APS patients were encompassed by 33 OAPS and 41 TAPS patients carefully selected for fulfilling the clinical and laboratory Sydney criteria [1] for APS classification. The APS patient enrolment was done in only one centre (Vall d’Hebron Hospital, Barcelona) because the use of a non-standardised methodology and design demanded data acquisition and analysis to be uniformly performed. In addition to describing a set of aPL that is unique for each clinical phenotype of APS, we found for those transiently seronegative OAPS and TAPS patients a positivity for non-criteria aPL detected by LIA and by ELISA. Moreover, this different set of aPL positivity suggests that distinct molecular mechanisms are involved in the pathology of APS and would require future research studies. Different aPL positivity due to gender and age is also a factor that requires attention. Our two groups of APS patients (OAPS and TAPS) presented a bias in gender and age composition and future research must take into account a balanced design. In our study, the group of healthy donors was gender- and age-matched against the whole APS patients.

## 4. Materials and Methods

### 4.1. Patients

Patients already diagnosed with APS (*n* = 74, 33 with OAPS and 41 with TAPS) were enrolled. Variable age was considered at enrolment. Blood collected by venipuncture was processed for serum isolation. A second blood sample was obtained from those patients attending medical consultation three months after the first withdrawal. Sera from age- and gender-matched healthy donors (*n* = 79) were obtained from the Catalan Biobank (Banc de Sang i Teixits, BST). Sera were collected from gel clot tubes (vacutainer, BD) after centrifugation for 15 min at 1500× *g*. Serum aliquots were stored at −80 °C. Clinical history and laboratory data of these APS patients were summarised in a codified and anonymised database and stored in our institutional repository (Hospital Universitari Vall d’Hebron-Institut de Recerca, HUVH-VHIR). All participants signed an informed consent form. This informed consent together with the research proposal project was approved by the local (HUVH-VHIR) ethics committee (PR(AG)83/2020).

### 4.2. Laboratory Clinics

Arterial hypertension was defined as blood pressure measure in the clinic ≥ 140/90 mm Hg in two or more assessments or when patient was taking oral anti-hypertensive medication. Serum total cholesterol levels were determined with standardised enzymatic methods by the clinical biochemistry laboratory of HUVH-VHIR and dyslipidaemia was defined as total cholesterol values ≥ 240 mg/dL. LA was measured at the clinical haematology laboratory of HUVH-VHIR by using standardised protocols of diluted Russell viper venom time (dRVVT) and silica clotting time (SCT). A patient was considered positive for LA when any test reached values ≥ 1.2. aCL IgG/IgM and aβ2GPI IgG/IgM were measured at the clinical immunology laboratory of HUVH-VHIR. They were assessed by commercial chemiluminescence assays (CLIA; QUANTA Flash, Werfen, Spain) with the BIO-FLASH system (Werfen, Spain). A patient was considered positive for aCL IgG/IgM when aCL CLIA values tested ≥ 40 GPL/MPL and positive for aβ2GPI IgG/IgM when aβ2GPI CLIA values were ≥40 AU. A patient fulfilling clinical criteria for APS according to Sydney criteria [1] and with any aPL positivity in two blood samples collected at least 12 weeks apart was considered as an APS patient. Depending on their aPL positivity, APS patients were classified by laboratory categories: category I, triple (positive for LA and aCL and aβ2GPI) or double (any combination of 2 positives) positivity; category II, single positivity for LA (IIa), aCL (IIb) or aβ2GPI (IIc).

### 4.3. LIA for the Detection aPL

Simultaneous detection of the lipid reactive antibodies aCL, aPE, aPS, aPA, aPC, aPG, aPI, as well as the phospholipid binding protein antibodies aβ2GPI, aA5, and aPT was performed in diluted (30 µL in 1 mL) serum samples following the recommendations of the manufacturer (GA Generic Assays GmbH, Dahlewtiz, Germany). LIA strips were scanned with the GelDoc XR and GelDoc XRplus (Bio-Rad, Hercules, CA, USA) instruments and densitometrically analysed with the Quantity One and Image Lab (Bio-Rad, Hercules, CA, USA) software. Background was subtracted for each sample. The resulting mean density value was considered as the aPL reactivity for each respective aPL and reported in arbitrary units (AU). aPL reactivity can also be deemed as aPL level. High positivity was considered when aPL reactivity reached the 99th percentile, assessed from the aPL reactivity of 79 serum samples from healthy donors. Medium positivity was considered when aPL reactivity reached the 95th percentile assessed as above.

### 4.4. Quantitative Linear Detection by LIA Technique

To study the linearity in the detection of aPL by LIA, serum diluted 1 in 33 (30 µL in 1 mL) from an aPL-positive donor showing positivity for aCL IgG (1584 GPL), aβ2GPI IgG (>6100 AU) and LA (dRVVTS = 3.96; dRVTTC = 2.44) was further serially diluted 0.5, 2, 4, 10, 50, 100 and 500 times. aPL detection was performed as described above.

### 4.5. Anti-Phosphatidylserine/Prothrombin ELISA

aPS/PT IgG and IgM were measured in 96 well microplates by ELISA (Aeskulisa Serine-Prothrombin-GM, AESKU.DIAGNOSTICS, Wendelsheim, Germany) following the procedures indicated by the manufacturer. Optical density was obtained by reading the microplates at 450 nm and 620 nm, for background subtraction, in a microplate reader (Bio Tek ELx800, Agilent, Santa Clara, CA, USA). Sera from 79 healthy donors were used to setup the cut-off at 99th percentile for aPS/PT IgG and IgM, which resulted in a value of 17 U/mL, nearly equal to the 18 U/mL established by the manufacturers. Patients above 18 U/mL were deemed as positive for aPS/PT. Those patients between 14 U/mL and 18 U/mL were considered as low–middle positive.

### 4.6. Statistical Analysis

Data were analysed using R software (version 4.2.1) and packages dplyr, tidyverse and R-stats. Data were grouped by OAPS versus TAPS. Continuous variables were summarised as means and standard deviations (SD) or medians and interquartile ranges (IQR). Categorical variables are presented as absolute number and relative frequency. Bivariate analyses were done by an unpaired t-test for continuous parametric variables or the Mann–Whitney U test for non-parametric variables. Assumption of normal distribution was proven with Shapiro–Wilk tests (*p* > 0.05 normally distributed data assumed) and Q-Q plots. Fisher’s exact test or the Chi-squared test were used for categorical variables. A two-sided α-level of 0.05 was deemed statistically significant. Multivariate analyses for principal component analysis and correlation were done in R with the packages FactoMineR and factoextra. Multivariate analyses of association between continuous variables were done in R package ggcorrplot.

## 5. Conclusions

A different profile of aPL is displayed by OAPS and TAPS patients by means of multiplexed LIA that combined criteria and non-criteria aPL. It suggests different molecular signalling elicited by the aPL would be the underlying pathological mechanisms for each APS phenotype. The dynamics of aPL positivity were observed in APS patients and rendered some of these patients as transiently seronegative APS patients. Non-criteria aPL positivity would help in the diagnosis of APS and the assessment of these aPL would prevent the underestimation of these patients.

## Figures and Tables

**Figure 1 ijms-23-12819-f001:**
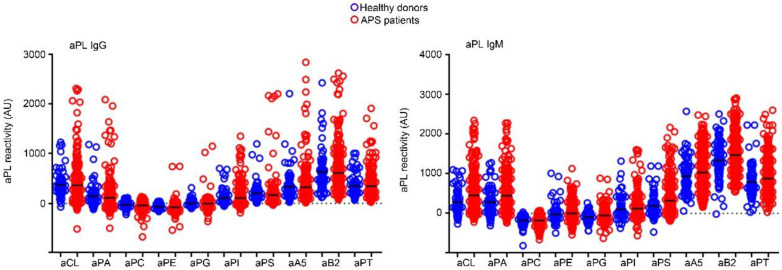
Antiphospholipid antibody (aPL) levels in sera of healthy individuals (blue dots) and antiphospholipid syndrome (APS) patients (red dots). Quantification of each aPL was run by densitometry of their respective bands on the strip of line immunoassay (LIA) incubated with anti-human IgG (left graph) or IgM (right graph). Net mean density values are reported as arbitrary units (AU) as indicated in Section 4.3. of Materials and Methods. Dark lines show the mean of each aPL’s reactivity. aCL, anti-cardiolipin; aPA, anti-phosphatidic acid; aPI, anti-phosphatidylinositol; aPC, anti-phosphatidylcholine; aPE, anti-phosphatidylethanolamine; aPG, anti-phosphatidylglycerol; aPS, anti-phosphatidylserine; aB2, anti-β2-glycoprotein-1; aA5, anti-annexin 5; aPT, anti-prothrombin.

**Figure 2 ijms-23-12819-f002:**
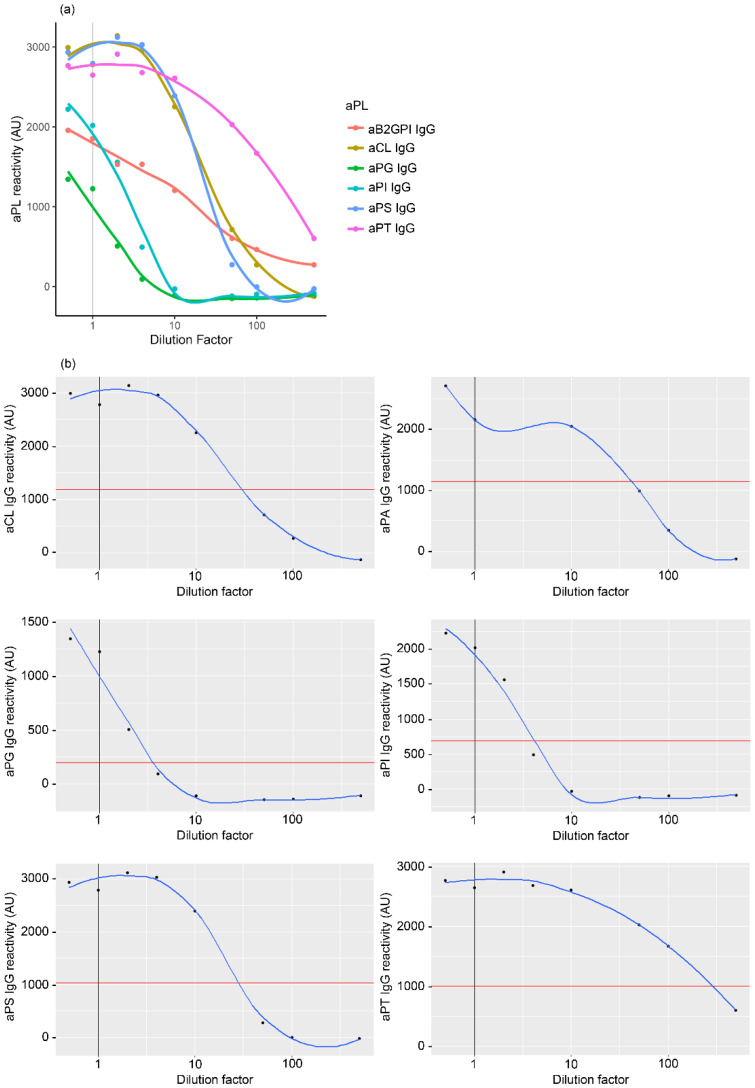
Quantification of the linear detection of antiphospholipid antibody (aPL). Serum from a triple-positive patient with antiphospholipid syndrome (APS) was serially diluted to assess aPL levels by line immunoassay (LIA). (**a**) Levels of different aPL were detected by LIA at the indicated dilutions. Dilution factor 1 (vertical grey line) designed the dilution recommended by the LIA manufacturer (1:33). The other dilution factors are in reference to this dilution factor 1. (**b**) Individual aPL levels for anti-cardiolipin (aCL) IgG (top left), anti-phosphatidic acid (aPA) IgG (top right), anti-phosphatidylglycerol (aPG) IgG (middle left), anti-phosphatidylinositol (aPI) IgG (middle right), anti-phosphatidylserine (aPS) IgG (bottom left), and anti-prothrombin (aPT) IgG (bottom right) at their respective dilution factors. Horizontal red line denotes the 99th-percentile value for each aPL. Experiments were performed in triplicate, showing the mean values of aPL reactivity reported in arbitrary units (AU) as indicated in Section 4.3 of Materials and Methods.

**Figure 3 ijms-23-12819-f003:**
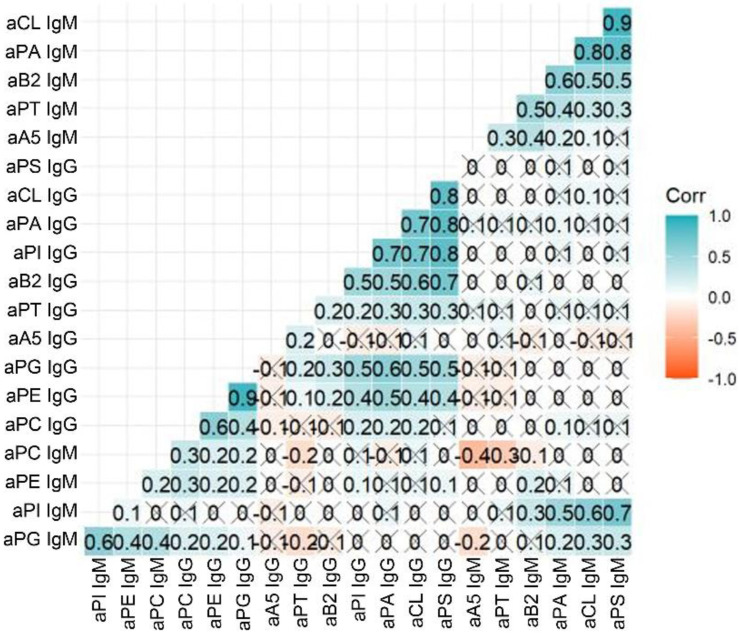
Antiphospholipid antibody (aPL) IgG and IgM associations. Multivariate analysis performed by R package ggcorrplot analysed the associations between the different aPL. aCL, anti-cardiolipin; aPA, anti-phosphatidic acid; aPI, anti-phosphatidylinositol; aPC, anti-phosphatidylcholine; aPE, anti-phosphatidylethanolamine; aPG, anti-phosphatidylglycerol; aPS, anti-phosphatidylserine; aB2, anti-β2-glycoprotein-1; aA5, anti-annexin 5; aPT, anti-prothrombin.

**Figure 4 ijms-23-12819-f004:**
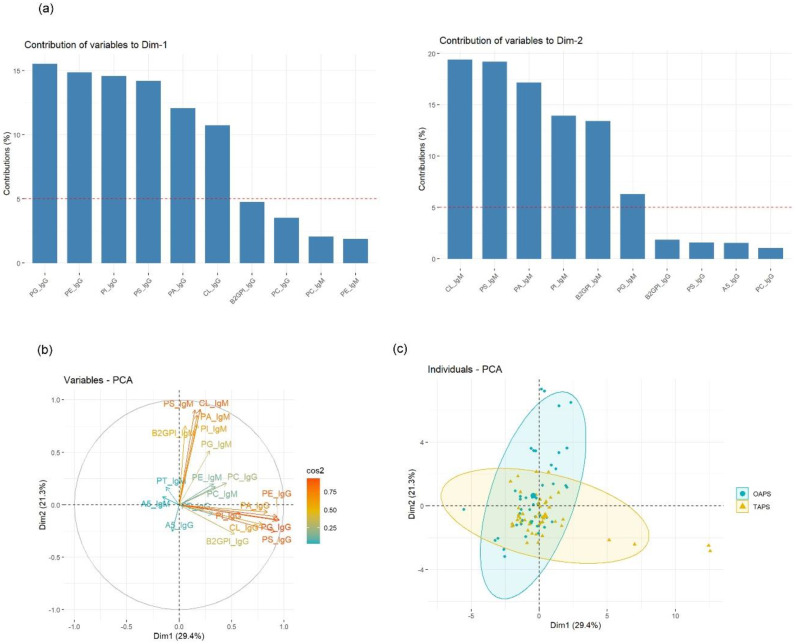
Antiphospholipid antibody (aPL) clustering for obstetric (OAPS) and thrombotic antiphospholipid syndrome (TAPS) patients. A principal component analysis (PCA) of logistic regression by using the R packages FactoMineR and factoextra was run. (**a**) The histograms show the selected variables and their value of contribution to define Dimension 1 (*x*-axis) and Dimension 2 (*y*-axis). (**b**) PCA bidimensional reduction resulting from using the defined Dimensions 1 and 2 and the degree of correlation (cos2) between the variables. (**c**) OAPS and TAPS patients plotted on this PCA based on their aPL levels. CL, anti-cardiolipin; PA, anti-phosphatidic acid; PI, anti-phosphatidylinositol; PC, anti-phosphatidylcholine; PE, anti-phosphatidylethanolamine; PG, anti-phosphatidylglycerol; PS, anti-phosphatidylserine; B2GPI, anti-β2-glycoprotein-1; A5, anti-annexin 5; PT, anti-prothrombin.

**Figure 5 ijms-23-12819-f005:**
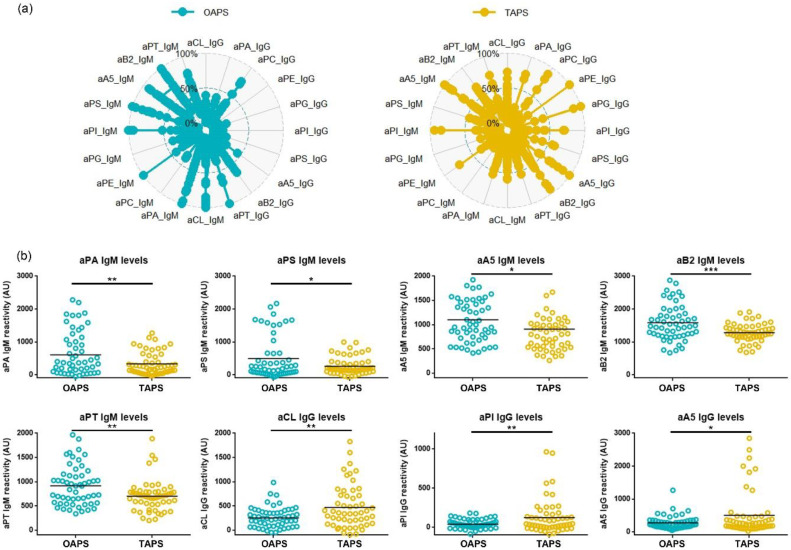
Different antiphospholipid antibody (aPL) levels between obstetric (OAPS) and thrombotic antiphospholipid syndrome (TAPS) patients. (**a**) Radar plots of individual aPL levels. Each aPL was normalised to its maximum of aPL reactivity reported in arbitrary units (AU) as indicated in Section 4.3 of Materials and Methods. (**b**) aPL that showed significant differences in the aPL levels (*t*-test, *p* < 0.05) between OAPS (blue dots) and TAPS (yellow dots) patients were selected. IgM of anti-phosphatidic acid (aPA), anti-phosphatidylserine (aPS), anti-annexin 5 (aA5), anti-β2 glycoprotein I (aβ2GPI) and anti-prothrombin (aPT) displayed higher levels in OAPS than in TAPS patients. IgG of aCL, anti-phosphatidylinositol (aPI) and aA5 presented higher levels in TAPS than in OAPS patients. * *p* < 0.05, ** *p* < 0.01, *** *p* < 0.001.

**Figure 6 ijms-23-12819-f006:**
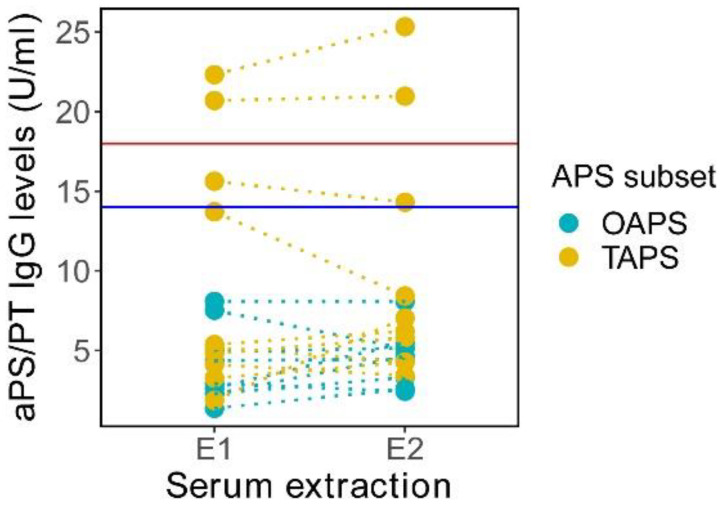
Antiphosphatidylserine/prothrombin (aPS/PT) IgG levels in transient seronegative obstetric (OAPS) and thrombotic antiphospholipid syndrome (TAPS) patients. Sera from transient seronegative OAPS and TAPS patients were isolated twice at least 12 weeks apart (E1 = extraction 1, E2 = extraction 2) and the IgG aPS/PT levels were measured by ELISA. Dotted lines represen both extractions of each patient. Horizontal red line denotes the threshold for aPS/PT positivity (>18 U/mL). Horizontal blue line denotes the limit for low-middle aPS/PT levels (>14 U/mL and <18 U/mL).

**Table 1 ijms-23-12819-t001:** Demographics of obstetric antiphospholipid syndrome (OAPS) and thrombotic antiphospholipid syndrome (TAPS) patients.

Feature	OAPS	TAPS	*p*-Value
Number	33	41	
Age (median (IQR))	39.96 (5.95)	53.44 (18.39)	<0.0001 *
Gender (Female (%))	33 (100)	16 (39.0)	<0.0001 **
BMI (median (IQR))	22.94 (4.31)	29.79 (7.47)	<0.0001 *
AHT (%)	3 (9.7)	22 (53.7)	<0.001 **
Dyslipidaemia (%)	0 (0)	22 (53.7)	<0.0001 **
AID (%)	4 (12.1)	7 (17.1)	0.746 **
Lab cat ^1^ (%):			0.072 ***
I	6 (18.2)	7 (17.1)	
IIa	18 (54.5)	30 (73.2)	
IIb	2 (6.1)	3 (7.3)	
IIc	7 (21.2)	1 (2.4)	
aPL persistency (%):			0.018 **
Persistent	9 (27.3)	23 (56.1)	
Oscillatory	24 (72.7)	18 (43.9)	

^1^ Laboratory category: category I, triple or doble positivity for any criteria aPL; category II, single positivity for LA (IIa), aCL (IIb), or aβ2GPI (IIc). AHT, arterial hypertension; aPL, antiphospholipid antibody; BMI, body mass index; Lab cat, laboratory category; AID, other autoimmune diseases. * Mann-Whitney U test; ** Fisher’s exact test; *** Pearson’s Chi-squared test.

**Table 2 ijms-23-12819-t002:** Percentiles for each antiphospholipid antibody (aPL) assessed by line immunoassay (LIA).

	aPL IgG								
Percentile	aCL	aPA	aPE	aPG	aPI	aPS	aA5	aβ2GPI	aPT
90th	708	355	2	65	287	353	629	1016	711
95th	880	562	12	104	504	567	828	1532	787
99th	1193	1148	44	199	688	1029	1421	1958	1006
	**aPL IgM**								
	**aCL**	**aPA**	**aPE**	**aPG**	**aPI**	**aPS**	**aA5**	**aβ2GPI**	**aPT**
90th	757	703	225	36	421	547	1480	1981	1231
95th	911	1054	455	72	657	767	1617	2115	1341
99th	1082	1260	953	183	1073	1198	1943	2375	2231

Antiphospholipid antibodies to cardiolipin (aCL), phosphatidic acid (aPA), phosphatidylinositol (aPI), phosphatidylcholine (aPC), phosphatidylethanolamine (aPE), phosphatidylglycerol (aPG), phosphatidylserine (aPS), β2-glycoprotein-1 (aβ2GPI), annexin 5 (aA5), and prothrombin (aPT) were determined by LIA. Values are the mean density arbitrary units resulting from the densitometry of aPL bands on each LIA strip.

**Table 3 ijms-23-12819-t003:** Percentage of obstetric antiphospholipid syndrome (OAPS) and thrombotic antiphospholipid syndrome (TAPS) patients who tested positive for the indicated antiphospholipid antibody (aPL).

aPL	OAPS	TAPS	*p*-Value *
aPA IgM	28.26	6.25	0.010
aPS IgM	23.91	6.25	0.034
aβ2GPI IgM	19.57	0	0.004
aPT IgM	17.39	4.17	0.081
aCL IgG	0	14.58	0.021
aPI IgG	0	6.25	0.256
aA5 IgG	0	16.57	0.012

aPA, anti-phosphatidic acid; aPS, anti-phosphatidylserine; aβ2GPI, anti-β2 Glycoprotein I; aPT, anti-prothrombin; aCL, anti-cardiolipin; aA5, anti-annexin 5. * Pearson’s Chi-squared test with Yates’ continuity correction.

**Table 4 ijms-23-12819-t004:** Percentage of currently seronegative patients in obstetric antiphospholipid syndrome (OAPS) and thrombotic antiphospholipid syndrome (TAPS) who tested positive for the non-criteria antiphospholipid antibody (nc aPL).

aPL	OAPS	TAPS	*p*-Value *
Seronegative ^1^ (%)	10 (30.3)	14 (34.1)	0.806
nc aPL pos ^2^ (%)	6 (60.0)	9 (64.3)	1
Twice nc aPL pos ^3^ (%)	2 (20.0)	8 (57.1)	0.104

^1^ APS diagnosed patients who at the moment of enrolment were seronegative in criteria aPL. ^2^ APS patients with positive results in any of the non-criteria antiphospholipid used in the line immunoassay. ^3^ APS patients with two positive results for the same non-criteria antiphospholipid used in the line immunoassay in serum samples drawn >12 weeks apart. * Fisher’s exact test.

## Data Availability

Data used for this study are stored in our institutional repository and will be available upon reasonable request to the corresponding authors.

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
