# Peer review of "Differences in Antiphospholipid Antibody Profile between Patients with Obstetric and Thrombotic Antiphospholipid Syndrome"

_ijms, 2022, doi:10.3390/ijms232112819_

Round 1

Reviewer 1 Report

Non-criteria APS is an interesting cohort of patients that needs better definition. The authors have used the LIA assay and do show clustering of aPLs that can differentiate between OAPS and TAPS. 

The drawbacks are the use of a single assay technique, given that there is no standardisation for the no-criteria assays. There is also no correlation with a differential effect on pathological signalling that underlies the phenotype. 

Overall, the number of patients is small, nevertheless the observations are interesting to the APS readership.

Author Response

Dear Editor and Reviewer

We have addressed all reviewers’ requests and amended the manuscript accordingly to their comments. We are grateful to the reviewers for their remarks which, we believe, change up the manuscript. We provide a detailed point-by-point reply and have made corresponding tracked changes in the attached manuscript.

Reviewer 1

Reviewer Q1: Non-criteria APS is an interesting cohort of patients that needs better definition. The authors have used the LIA assay and do show clustering of aPLs that can differentiate between OAPS and TAPS.

Answer Q1: We thank the reviewer for this summarised take-home message.

Reviewer Q2: The drawbacks are the use of a single assay technique, given that there is no standardisation for the no-criteria assays. There is also no correlation with a differential effect on pathological signalling that underlies the phenotype. 

Answer Q2: As the reviewer pointed out, it is difficult to find standardized methodologies for testing non-criteria aPL. We choose the line immunoassay (LIA) already reported in the APS literature. With this multiplex screening we can check for the presence of 10 different IgG or IgM aPL on sera from APS individuals. Although this methodology is not yet standardized and is not currently used in CORE clinic labs our study contributes to the implementation of this technique as it was previously done for the definition of thrombotic risk profiles (Park et al. 2019 Clin Lab; 65: 207-213). Moreover, we have used a second non-criteria aPL assay, the anti-phosphatidylserine/prothrombin ELISA. The new aPS/PT ELISA data were incorporated in results and discussion sections and is pasted below.

We obtained a profile of aPL that is different between thrombotic and obstetric APS, and the implication of molecular pathological mechanisms that underlies each phenotype was suggested and will be explored in future researches.

The new text in results is:

“2.7. non-criteria aPL positivity in transient seronegative TAPS and OAPS patients

We previously mentioned that some OAPS and TAPS patients were seronegative at the moment of enrolment. We analysed in these transient seronegative APS patients (10 OAPS and 14 TAPS) for the presence of non-criteria aPL (Table 4). The number of APS patients who resulted positive for any of the non-criteria aPL tested in the LIA was 6 out of 10 (60.0%) in OAPS and 9 out of 14 (64.3%) in TAPS patients. Moreover, we drew blood from these patients in a lapse of time greater than 12 weeks and tested their sera again by LIA. The number of patients with twice medium (>95th percentile) or high (>99th percentile) titers for the same non-criteria aPL were 2 (20.0 %) in OAPS and 8 (57.1%) in TAPS. There were no differences in the positivity of non-criteria aPL between seronegative OAPS and TAPS (Table 4). The non-criteria antibody that tested twice positive in the seronegative OAPS were aA5, aPE, aPG, and aPA. We found an enlarged list of the non-criteria aPL that tested twice positive in seronegative TAPS: aA5, aPE, aPG, aPA, aPI, aPS and aPT.

Since three TAPS patients showed twice aPS and/or aPT positivity, we decided to test all the seronegative APS patients for aPS/PT by ELISA. We observed that none of the seronegative OAPS patients yielded positive for aPS/PT. The two seronegative TAPS patients who were positive for aPT by LIA also tested positive for aPS/PT by ELISA. A third TAPS patient with twice aPS positivity resulted in medium positivity by aPS/PT (Figure 6).”

And in discussion:

“Our data described that thrombotic APS patients who transiently tested seronegative for criteria aPL, could be detected twice positive for aPS and/or aPT by two different techniques (aPS, aPT by LIA, and aPS/PT by ELISA).”

Reviewer Q3: Overall, the number of patients is small, nevertheless the observations are interesting to the APS readership.

Answer Q3: The reviewer is right on the final number of patients for each subset of APS. This observation is now included as one of the limitations of the study in the discussion that reads:

“A limitation of the study is the relatively small number of OAPS and TAPS patients. Nevertheless, we tested multiplex LIA in a total of 153 individuals, whose 79 were healthy donors and 74 APS patients. Data from healthy donors was used to define the cut-off for each aPL on the multiplex assay. APS patients were encompassed by 33 OAPS and 41 TAPS patients carefully selected for fulfilling the clinical and laboratory Sydney criteria [1] for APS classification. The APS patient enrolment was done in only one centre (Vall d’Hebron Hospital, Barcelona) because the use of a non-standardised methodology and design demanded data acquisition and analysis uniformly performed.”

Reviewer 2 Report

In this study the authors showed differences in the aPL profile between patients presenting a thrombotic and obstetrical antiphospholipid syndrome.

This question this question has already been considered in other publications with controversial results. In this study, the authors studied a very small number of patients. Several aPLs have been researched, some of which are of no research interest in current practice. No clear conclusion is therefore possible.

Author Response

We have addressed all reviewers’ requests and amended the manuscript accordingly to their comments. We are grateful to the reviewers for their remarks which, we believe, change up the manuscript. We provide a detailed point-by-point reply and have made corresponding tracked changes in the attached manuscript.

Reviewer 2

Reviewer Q1: In this study the authors showed differences in the aPL profile between patients presenting a thrombotic and obstetrical antiphospholipid syndrome. This question has already been considered in other publications with controversial results.

Answer Q1: To the best of our knowledge the use of multiplexed line immunoassay for APS profiling by comparing TAPS and OAPS patients was reported by us in a Letter to the Editor in Arthritis & Rheumatism journal (already referenced in the introduction of our manuscript [16]). In that occasion 45 APS patients including 3 with obstetric morbidities were analysed. Besides this small OAPS group, the data were not quantitatively analysed and therefore, multivariate analysis that allows clustering based on multiple data of aPL levels were not performed.

Reviewer Q2: In this study, the authors studied a very small number of patients.

Answer Q2: This point was also remarked by reviewer 1 and we must agree with both reviewers about the apparent small number of APS patients. For more than two years we attended and enrolled after a written consent a total of 250 putative APS patients apart from 79 healthy donors. After careful assessment of clinical history and laboratory data, the remaining number of patients that could be clearly classified as APS patients according to Sydney classification criteria was 74 (33 OAPS and 41 TAPS). Our database, that is an example of how difficult is to obtain a significant number of APS patients will be made available to the editors whether they consider it is required.

The small number of patients analysed in this study is deemed as a limitation and we added the following text to the discussion:

“A limitation of the study is the relatively small number of OAPS and TAPS patients. Nevertheless, we tested multiplex LIA in a total of 153 individuals, whose 79 were healthy donors and 74 APS patients. Data from healthy donors was used to define the cut-off for each aPL on the multiplex assay. APS patients were encompassed by 33 OAPS and 41 TAPS patients carefully selected for fulfilling the clinical and laboratory Sydney criteria [1] for APS classification. The APS patient enrolment was done in only one centre (Vall d’Hebron Hospital, Barcelona) because the use of a non-standardised methodology and design demanded data acquisition and analysis uniformly performed.”

Reviewer Q3: Several aPLs have been researched, some of which are of no research interest in current practice. No clear conclusion is therefore possible.

Answer Q3: The reviewer is right about how confusing are the data in the literature on the meaning of aPL positivity for several aPL. This confusion may cause discouragement in the current practice. That is why we hardly believe that performing basic research will help in the definition of significant aPL that could have a role in the pathogenesis of currently seronegative APS patients. In this study we included already diagnosed APS patients, that showed twice positivity for LA, aCL or aβ2GPI within the five years close to the clinical APS criteria. Nevertheless, aPL titres are changing and some of these individuals become seronegative whilst retaining clinical symptomatology. Presence of underlying aPL were tested by LIA. Positivity for each of the aPL encompassed in the multiplexed LIA was established by testing 79 healthy donors. Our study is describing an overall profile of aPL reactivity in OAPS and TAPS patients, but also is describing some individual aPL positivity including non-criteria aPL. Of note, we do not know the pathological molecular mechanisms engaged by the presence of a particular aPL positivity, but this is a step by step process of research we are engaged in. We have added new data about the advantage of using two techniques (LIA and aPS/PT ELISA) in order to recover those temporally seronegative APS but with some non-criteria aPL positivity. The text incorporated in results section is:

“2.7. non-criteria aPL positivity in transient seronegative TAPS and OAPS patients

We previously mentioned that some OAPS and TAPS patients were seronegative at the moment of enrolment. We analysed in these transient seronegative APS patients (10 OAPS and 14 TAPS) for the presence of non-criteria aPL (Table 4). The number of APS patients who resulted positive for any of the non-criteria aPL tested in the LIA was 6 out of 10 (60.0%) in OAPS and 9 out of 14 (64.3%) in TAPS patients. Moreover, we drew blood from these patients in a lapse of time greater than 12 weeks and tested their sera again by LIA. The number of patients with twice medium (>95th percentile) or high (>99th percentile) titers for the same non-criteria aPL were 2 (20.0 %) in OAPS and 8 (57.1%) in TAPS. There were no differences in the positivity of non-criteria aPL between seronegative OAPS and TAPS (Table 4). The non-criteria antibody that tested twice positive in the seronegative OAPS were aA5, aPE, aPG, and aPA. We found an enlarged list of the non-criteria aPL that tested twice positive in seronegative TAPS: aA5, aPE, aPG, aPA, aPI, aPS and aPT.

Since three TAPS patients showed twice aPS and/or aPT positivity, we decided to test all the seronegative APS patients for aPS/PT by ELISA. We observed that none of the seronegative OAPS patients yielded positive for aPS/PT. The two seronegative TAPS patients who were positive for aPT by LIA also tested positive for aPS/PT by ELISA. A third TAPS patient with twice aPS positivity resulted in medium positivity by aPS/PT (Figure 6).”

Reviewer 3 Report

Reviewing of the article entitled:

Differences in Antiphospholipid Antibody Profile between Patients with Obstetric and Thrombotic Antiphospholipid Syndrome

Submitted to the International Journal of Molecular Sciences

Abstract:

The abstract is hard to read and to understand as a first approach of your work.

To help clarifying your topic, please :

1)     Define your population of patients: how many were patients without criteria aPL or seronegative APS in each group? What percentage of women did you have in the thrombotic APS group (it matters, as you compared thrombotic APS patients with obstetric APS patients who are, by definition, only women)?

2)     Define all the aPL antibodies that you measured, and distinguish criteria aPL from non-criteria aPL.

3)     Simplify the description of your results. It is too complicated and too long. We need a simple take-home message.

Introduction:

It is very clear and very interesting, thank you. You define well the current knowledge on your topic, and explain well the need to investigate non criteria aPL, and to characterize better seronegative APS patients.

Material and methods: the paragraph on Material and methods must be BEFORE the results!

How did you define seronegative thrombotic APS and seronegative obstetric APS? Indeed, vasculo-placental disorders are not specific for APS, so how did you distinguish those women from non APS women?

Results:

-        Rephrase : “Patients were split into obstetric (OAPS, n=33) and thrombotic (TAPS, n=41)”. It is not proper.

-        Did you verify that the different clusters were not related to gender? This is a big issue with gender in your work. Differences in aPL positivity between thrombotic APS and obstetric APS could only be due to gender.

-        Figure 5: can you add on your graphs p values with symbols like “*”?

-        Table 3: can you add the results according to the 2 subgroups of patients in each group: APS patients with aPL criteria and seronegative APS patients? Your introduction explains well the interest of exploring seronegative APS patients, but you do not study their results separately, so it is unfortunate and we are not able to connect the ideas of your introduction with the presentation of your results.

Conclusion: you need to add a paragraph with a conclusion at the end of your manuscript.

Author Response

Dear Editor and Reviewer

We have addressed all reviewers’ requests and amended the manuscript accordingly to their comments. We are grateful to the reviewers for their remarks which, we believe, change up the manuscript. We provide a detailed point-by-point reply and have made corresponding tracked changes in the attached manuscript.

Reviewer 3

Reviewer Q1: Abstract: The abstract is hard to read and to understand as a first approach of your work.

Answer Q1: We hardly appreciate the comments made by the reviewer as we thank his/her advises. We addressed every request by making the indicated changes. Thus, we edited a new abstract that we believe it was improved thanks to the reviewer recommendations. Please see the addressed points below. 

Reviewer Q2: 1) Define your population of patients: how many were patients without criteria aPL or seronegative APS in each group? What percentage of women did you have in the thrombotic APS group (it matters, as you compared thrombotic APS patients with obstetric APS patients who are, by definition, only women)?

Answer Q2: All the APS patients were fulfilling the Sydney laboratory criteria and were positive twice for lupus anticoagulant, anticardiolipin and/or anti-β2glicoprotein I antibodies within the 5 years of clinical APS event. Nevertheless, some APS patients showed an oscillatory aPL detection (42 out of 74) and 24 (32.4 %) were seronegative at enrolment. Of these 74 APS individuals, 33 were obstetric APS (OAPS) and 41 thrombotic APS (TAPS). 39% of TAPS patients were women. Now the definition of our population reads:

“In a cohort of 74 yet diagnosed APS individuals fulfilling Sydney laboratory criteria (twice positive for lupus anticoagulant, anticardiolipin, aCL, and/or anti-β2glycoprotein I, aβ2GPI), 33 out of 74 were obstetric APS (OAPS) and 41 thrombotic APS (TAPS). 39% of TAPS patients were women. Although aPL detection was persistent, we observed an oscillatory aPL positivity in 56,7% and a transient seroconversion in 32,4% of APS patients at enrolment”

Reviewer Q3: 2) Define all the aPL antibodies that you measured, and distinguish criteria aPL from non-criteria aPL.

Answer Q3: We defined all the aPL as follows:

“Thus, we tested their sera in a line immunoassay that simultaneously detected criteria (aCL and aβ2GPI) and non-criteria (anti-phosphatidylserine, aPS; anti-phosphatidic acid, aPA; anti-phosphatidylinositol, aPI; anti-annexin 5, aA5; anti-prothrombin, aPT; anti-phosphatidylethanolamine; anti-phosphatidylglycerol, and anti-phosphatidylcholine) aPL.”

Reviewer Q4: 3) Simplify the description of your results. It is too complicated and too long. We need a simple take-home message.

Answer Q4: The description of our results now reads:

“OAPS and TAPS patients displayed different but overlapping clusters based on their aPL reactivities. Specifically, while OAPS patients showed higher aPA, aPS, aA5, aβ2GPI and aPT IgM levels than TAPS patients, the latter displayed higher reactivity in aCL, aPI and aA5 IgG. Eventually, with a cut-off of 99th-percentile stablished from a population of 79 healthy donors, TAPS patients significantly tested more positive for aCL and aA5 IgG than OAPS patients who tested more positive for aPA, aPS and aβ2GPI IgM. Transiently seronegative APS patients showed non-criteria aPL positivity twice in sera obtained 3 months apart”.

And our take home-message is:

“Overall, our data show that APS patients presented clusters of aPL that define different profile between OAPS and TAPS, and persistent non-criteria aPL positivity was observed in those who are transiently seronegative”

Reviewer Q5: Introduction: It is very clear and very interesting, thank you. You define well the current knowledge on your topic, and explain well the need to investigate non criteria aPL, and to characterize better seronegative APS patients.

Answer Q5: We thank the reviewer for this kind assessment of the introduction

Reviewer Q6: Material and methods: the paragraph on Material and methods must be BEFORE the results!

Answer Q6: Our initial manuscript was considering this logical structure proposed by the reviewer. But the specific IJMS journal instructions place the Material and Methods section just after the Discussion. On the other hand, we added the aPS/PT ELISA method:

“4.6. anti-phosphatidylserine/prothrombin ELISA

                aPS/PT IgG and IgM were measured in 96 well microplates by ELISA (Aeskulisa Serine-Prothrombin-GM, AESKU.DIAGNOSTICS, Germany) following the procedures indicated by the manufacturer. Optical density was obtained by reading the microplates at 450nm and 620nm, for background subtraction, in a microplate reader (Bio Tek ELx800, Agilent, CA, USA). Sera from 79 healthy donors was used to setup the cut-off at 99th percentile for aPS/PT IgG and IgM which resulted at 17 U/ml, nearly equal to the 18 U/ml established by the manufacturers. Patients above 18 U/ml were deemed as positive for aPS/PT. Those patients between 14 U/ml and 18 U/ml were considered as low-middle positive.”

Reviewer Q6: How did you define seronegative thrombotic APS and seronegative obstetric APS? Indeed, vasculo-placental disorders are not specific for APS, so how did you distinguish those women from non APS women?

Answer Q6: All the enrolled APS patients (TAPS and OAPS) fulfilled the clinical as well as the laboratory Sydney criteria and were twice positive for lupus anticoagulant, anticardiolipin and/or anti-β2glicoprotein I within the 5 years of clinical APS event. Nevertheless, some APS patients showed an oscillatory aPL detection during the follow up and some of them temporally lose their aPL positivity.

Reviewer Q7: Results: Rephrase : “Patients were split into obstetric (OAPS, n=33) and thrombotic (TAPS, n=41)”. It is not proper.

Answer Q7: We have written the above sentence and now it reads:

“A cohort of 74 yet diagnosed APS patients fulfilling the laboratory Sydney criteria, twice positivity for LA, aCL and/or aβ2GPI, were divided according to their clinical criteria into obstetric (OAPS, n=33) and thrombotic (TAPS, n=41) APS patients”

Reviewer Q8: Did you verify that the different clusters were not related to gender? This is a big issue with gender in your work. Differences in aPL positivity between thrombotic APS and obstetric APS could only be due to gender.

Answer Q8: We totally agree that gender could play a role in clustering, and most importantly, women were less present in our subset of TAPS patients, only 39% of TAPS were women. To address this important issue, we performed the cluster analysis by selecting TAPS women and compared to the OAPS women. The resulting clustering was less performant due to less number of total APS patients, but it rendered the same tendency in OAPS vs TAPS cluster as we observed with all the APS patients. An image of this cluster is included in the attached word file and we added a comment on this limitation in the discussion that reads:

“Different aPL positivity due to gender and age is also a factor that requires attention. Our two groups of APS patients (OAPS and TAPS) presented a bias in gender and age composition and future research must take into account a balanced design. In our study the group of healthy donors was gender and age matched against the whole APS patients.”

Reviewer Q9: Figure 5: can you add on your graphs p values with symbols like “*”?

Answer Q9: We have added the symbols of p-values in the Figure 5 as well as their meaning in the legend: “* p<0.05, ** p<0.01, *** p<0.001”.

Reviewer Q10: Table 3: can you add the results according to the 2 subgroups of patients in each group: APS patients with aPL criteria and seronegative APS patients? Your introduction explains well the interest of exploring seronegative APS patients, but you do not study their results separately, so it is unfortunate and we are not able to connect the ideas of your introduction with the presentation of your results.

Answer Q10: We thank the reviewer for this important observation. We have rethought the data on the seronegative APS patients and presented it as new data on result section (please see the new figure 6 in the attached word file):

“2.7. non-criteria aPL positivity in transient seronegative TAPS and OAPS patients

We previously mentioned that some OAPS and TAPS patients were seronegative at the moment of enrolment. We analysed in these transient seronegative APS patients (10 OAPS and 14 TAPS) for the presence of non-criteria aPL (Table 4). The number of APS patients who resulted positive for any of the non-criteria aPL tested in the LIA was 6 out of 10 (60.0%) in OAPS and 9 out of 14 (64.3%) in TAPS patients. Moreover, we drew blood from these patients in a lapse of time greater than 12 weeks and tested their sera again by LIA. The number of patients with twice medium (>95th percentile) or high (>99th percentile) titers for the same non-criteria aPL were 2 (20.0 %) in OAPS and 8 (57.1%) in TAPS. There were no differences in the positivity of non-criteria aPL between seronegative OAPS and TAPS (Table 4). The non-criteria antibody that tested twice positive in the seronegative OAPS were aA5, aPE, aPG, and aPA. We found an enlarged list of the non-criteria aPL that tested twice positive in seronegative TAPS: aA5, aPE, aPG, aPA, aPI, aPS and aPT.

Since three TAPS patients showed twice aPS and/or aPT positivity, we decided to test all the seronegative APS patients for aPS/PT by ELISA. We observed that none of the seronegative OAPS patients yielded positive for aPS/PT. The two seronegative TAPS patients who were positive for aPT by LIA also tested positive for aPS/PT by ELISA. A third TAPS patient with twice aPS positivity resulted in medium positivity by aPS/PT (Figure 6).    

Table 4. Percentage of currently seronegative patients in obstetric antiphospholipid syndrome (OAPS) and thrombotic antiphospholipid syndrome (TAPS) who tested positive for the non-criteria antiphospholipid antibody (nc aPL).

aPL

OAPS

TAPS

p-value*

Seronegative1 (%)

10 (30.3)

14 (34.1)

0.806

nc aPL pos2 (%)

6 (60.0)

9 (64.3)

1

Twice nc aPL pos3 (%)

2 (20.0)

8 (57.1)

0.104

1 APS diagnosed patients who at the moment of enrolment were seronegative in criteria aPL. 2 APS patients with positive results in any of the non-criteria antiphospholipid used in the line immunoassay. 3 APS patients with two positive results for the same non-criteria antiphospholipid used in the line immunoassay in serum samples drawn >12 weeks apart. * Fisher’s exact test.

Figure 6. antiphosphatidylserine/prothrombin (aPS/PT) IgG levels in transient seronegative obstetric (OAPS) and thrombotic antiphospholipid syndrome (TAPS). Sera from transient seronegative OAPS and TAPS patients were isolated twice at least 12 weeks apart (E1= extraction 1, E2=extraction 2) and the IgG aPS/PT levels were measured by ELISA. Dotted lines binds both extractions of each patient. Horizontal red line denotes the threshold for aPS/PT positivity (>18U/ml). Horizontal blue line denotes the limit for low-middle aPS/PT levels (>14 U/ml and < 18 U/ml).”

And in the discussion section:

“the dynamics of aPL levels is also a matter of debate and many APS patients showed an oscillatory aPL positivity. Our data described that thrombotic APS patients who transiently tested seronegative for criteria aPL, could be detected twice positive for aPS and/or aPT by two different techniques (aPS, aPT by LIA, and aPS/PT by ELISA). Consequently, the reliability of APS diagnosis would improve with testing more aPL.”

Reviewer Q11: Conclusion: you need to add a paragraph with a conclusion at the end of your manuscript.

Answer Q11: A conclusion is added and it reads:

“5. Conclusions

A different profile of aPL is displayed by OAPS and TAPS patients by means of multiplexed LIA that combined criteria and non-criteria aPL. It suggests different molecular signalling elicited by the aPL would be the underlying pathological mechanisms for each APS phenotype. Dynamics of aPL positivity is observed in APS patients and render some of these patients as transiently seronegative APS. Non-criteria aPL positivity would help in the diagnosis of APS and assessment of these aPL would prevent the underestimation of these patients.”

We would like to thank the reviewer for his/her comments that clearly improved our manuscript

Round 2

Reviewer 2 Report

Reviewer Q1: My question was not about technical approach but concerned the distinction between thrombotic or obstetrical profile in antiphospholipid syndrome. This question has already been considered in other publications. As an example recently urinary CXCL12 and PDGFB were proposed as potential non-invasive markers to differentiate primary TAPS from primary OAPS (Front Immunol 2021 Aug 19;12:702425), underlying the fact that thrombotic and obstetrical clinical profile could be associated to two distinct pathologies. In this study no original result was found to corroborate this concept.

Reviewer Q2: As observed by the authors, the small number of patients analysed in this study is a huge limitation.

Reviewer Q3The question is what is the interest of the study, with a small population, and with markers which have already been studied without conclusive results.

Reviewer 3 Report

I thank the authors for the modifications made to their manuscript, in accordance with my previous comments.

These modifications improved much the manuscript and made it easier to understand for the readers.

It can now be published in the present form.